# Ostrich eggshell beads from Ga-Mohana Hill North Rockshelter, southern Kalahari, and the implications for understanding social networks during Marine Isotope Stage 2

Amy Hatton [1,2,3] *, Benjamin Collins [3,4], Benjamin J. Schoville [3,5], Jayne Wilkins [3,6]

1 Extreme Events Research Group, Max Planck Institutes for Science of Human History, Chemical Ecology and Biogeochemistry, Jena, Germany, 2 Department of Archaeology, Max Planck Institute for Science of Human History, Jena, Germany, 3 Department of Archaeology, University of Cape Town, Human Evolution Research Institute, Cape Town, Private Bag, Rondebosch, South Africa, 4 Department of Anthropology, University of Manitoba, Winnipeg, Manitoba, Canada, 5 School of Social Science, University of Queensland, St Lucia, Queensland, Australia, 6 Australian Research Centre for Human Evolution, Griffith University, Nathan, QLD, Australia

* hatton@shh.mpg.de

**Data Availability Statement:** All relevant data are within the paper and its Supporting Information

## Abstract

Ostrich eggshell (OES) beads from southern African archaeological contexts shed light on past traditions of personal ornamentation, and they are also argued to provide a proxy for understanding past social networks. However, OES beads are often understudied and not reported on in detail. In particular, there has been little research on OES bead variation during Marine Isotope Stage 2 (29,000–12,000 years ago) which includes the Last Glacial Maximum when changing climatic conditions are hypothesized to have significant impact on forager social networks. Here, we present the first technological analysis of terminal Pleistocene OES beads and fragments in the Kalahari from the ~15 ka levels at Ga-Mohana Hill North Rockshelter. We contextualise these findings through comparison with coeval OES bead assemblages across southern Africa during MIS 2. Results indicate that OES beads were manufactured at Ga-Mohana Hill North during the terminal Pleistocene occupation, based on the presence of most stages of bead manufacture. The review shows that OES beads were present across southern Africa through MIS 2, suggesting that culturing of the body was an embodied and persistent practice during that time. While the importance of OES beads as decorative objects was shared by populations across southern Africa, variation in bead diameters indicate that there was stylistic variation.

## Introduction

People, past and present, use jewellery, such as personal ornaments, to culture their bodies and communicate information about themselves both within and between groups. Beadwork characteristics, such as material, association, size, and location of beadworks embed messages of cultural and social significance [1,2]. The ability to create, manipulate, and communicate

files. Additionally, in order to ensure full reproducibility, source data and R scripts are available on a GitHub repository (https://github.com/amyhatton/ghn_bead_paper).

**Funding:** This research was funded by an Australian Research Council Discovery Early Career Research Award (DE 190100160) to JW, and a National Research Foundation (South Africa) Research Development Grant for Y-rated Researchers (116349) to JW. BJS received a Centre of Excellence in Palaeoscience Postdoctoral Fellowship and is supported by the University of Queensland Anthropocene Project.

**Competing interests:** The authors have declared that no competing interests exist.

through symbols is argued to be critical for developing and maintaining social connections within and between groups, which confers adaptive advantages in terms of social networks and potential safety nets [3–6].

Forager groups from the Kalahari Basin have provided insight into the manufacture of ostrich eggshell (OES) beads in modern ethnographic contexts [7,8]. These studies inform a range of approaches for interpreting the manufacture of OES beads, as well as their importance within social and cultural contexts in the past [9,10]. While these analogies are essential to our understanding of OES beads and their social context, one must be wary of how far back they can be extended. The analogies are based on ethnographic data from a small number of modern forager groups, mostly restricted to the Kalahari. Modern forager groups in southern Africa have been influenced by interaction and integration in shifting socio-political landscapes [11]. While practices such as OES bead manufacture have existed for thousands of years the uses and social practices surrounding these objects will have changed.

OES beads and fragments are rarely studied in detail, despite how common they are in Holocene contexts. Jacobson was the first to note differences in the sizes of OES beads between the archaeological assemblages of forager and pastoralists groups, where foragers generally manufactured smaller beads [12]. In recent years there has been more research interest in OES beads, with further studies examining the variation in bead size during the Holocene [13] and Late Pleistocene [14] across both eastern and southern Africa. Isotopic analyses of OES beads in Lesotho and the Maloti Drakensberg, showing that OES beads were traded across large distances (>300km) as far back as 33,000 years ago (ka) [5], while stylistic comparisons suggest social networks spanning eastern and southern Africa as far back as 55,000 years ago [14].

To date, there has been little research examining the variation in OES beads made by forager groups in Marine Isotope Stage 2 (MIS 2) (29–12 ka). During MIS 2, southern Africa experienced hypervariable climatic conditions, with many intervals that were overall cooler and drier than today, and may have presented unique environmental challenges to past foragers. Due to these conditions, MIS 2 has been hypothesized to represent a period of social coalescence and connectedness, especially after 22 ka, with the spread of 'Robberg'-designated assemblages [15].

The recovery of OES from archaeological contexts in the semi-arid Kalahari Basin is relatively rare. Previous studies of OES in the Kalahari are relatively narrow in scope and include analysis of basic characteristics such as the mean diameter of beads or the number and weight of OES fragments [16]. Here, we provide the first detailed, technological study of OES beads from a MIS 2 context in this region at the site of Ga-Mohana Hill North Rockshelter (GHN). We further contextualise this assemblage through comparisons with coeval OES bead assemblages across southern Africa. These data are complemented with the abundances of OES fragments, where available, to provide a more nuanced understanding of the use of ostrich eggs and OES bead manufacture.

## Ostrich eggshell beads

OES beads first appear in the archaeological record 50–40 ka in eastern and southern Africa and shortly thereafter in China [17–21]. Beads from southern Africa tend to generally be younger than their eastern African counterparts, with the oldest OES beads from southern Africa dating to 44–41 ka at Border Cave [18]. At Spitzkloof, two bead preforms were recovered from deposits dated to > 51 ka, but as isolated finds they remain difficult to interpret [22,23]. OES bead assemblages become ubiquitous in the Holocene, with both the size of assemblages and number of sites demonstrating OES beads increasing in magnitude in comparison to the

Pleistocene [13]. This increase may be linked to preservation issues, broader uptake of OES beads as decorations, or a combination of both.

Research of OES beads in archaeological contexts generally focuses on identifying variation in bead style, specifically changes or differences in OES bead technological properties, such as shape and diameter, and using these differences as proxies for inferring stylistic differences between cultural groups [12–14,18]. More recently, isotopic analyses have also been used to track the movement of OES beads across the landscape [5,24], with the results informing our understanding of the nature and extent of past exchange networks.

Jewellery comprised of OES beads are popular exchange items in! *hxaro*, a custom of exchange and gift-giving among Kalahari San populations, who traditionally practiced a foraging lifestyle [7,8,25]. Within the context of! *hxaro*, OES beads may travel hundreds of kilometres, with the gifts helping to initiate and maintain social bonds between different groups and disperse access rights to landscapes and resources [26,27]. There are acknowledged issues with direct ethnographic comparisons between recent and Palaeolithic foragers [11,28,29] however, the movement of OES beads across distances of >300 km for at least the past 30 ka, suggests that they may provide a cautious heuristic for interpreting the nature and extent of social networks in the past [5,30].

While OES beads have a long history of study, especially in southern Africa [12,31], OES fragments have not received the same level of interest. Data on OES fragments are often presented minimally, such as in summaries of the number of fragments or possibly the weight of fragments reported. While this is understandable given that there could be thousands of OES fragments at a given site, OES fragments are intrinsic to our understanding of how beads were made and traded among people through time, especially when considering where they were manufactured and the potential social and environmental constraints of their production [30]. Studies that do focus on OES fragments tend to describe those that were modified by past peoples, such as the engraved fragments from Diepkloof [32].

## Marine Isotope Stage 2

Marine Isotope Stages represent a summary of changes in global temperatures and sea-levels based on deep sea sediment cores [33]. MIS with odd numbers represent warmer interglacial periods with high sea-levels, and MIS with even numbers represent cooler glacial periods with low sea-levels. Archaeologists in southern African often use these MIS as temporal frameworks for the archaeological record [15,34–36]. MIS 2 is a glacial period from 29 to 12 ka, associated with the Last Glacial Maximum (LGM, conservatively dated to ~24–18 ka), when sea levels were globally an average of 125 m lower than they are today [37,38]. As a generalization, glacial periods in Africa are characterized by drier conditions that are often thought to pose resource challenges for foragers [39,40]. However, palaeoenvironmental analyses in some regions show the opposite of that general expectation [41]. In southern Africa, palaeoenvironmental records show cool temperatures during the LGM, with warming commencing about ~17 ka, then followed by a cool period ~13 to 11 ka [42]. The effect of these temperature changes on precipitation varied across southern Africa. Based on the compilation of several proxy palaeoenvironmental archives [42], the LGM was generally drier than it is today in the east and along the south coast, but wetter in the west along the coast and into the interior. However, this general pattern is not supported at all locales [43], attesting to the importance of investigating palaeoenvironments at the local scale.

The beginning of MIS 2 correlates roughly with a shift from Middle Stone Age (MSA) to Later Stone Age (LSA) technologies; lithic assemblages characterized by points and prepared cores shift to assemblages characterized by bladelets, backed pieces, and bipolar technology.

Many LSA assemblages reflect miniaturization, which is an emphasis on the production of pieces < 3–5 cm in size, potentially a response to hafting needs and/or raw material conservation [44]. However, at some sites, this shift occurred much earlier than MIS 2 [45] and the technological shift was not synchronous across southern Africa. Two technocomplexes are associated with MIS 2; the 'Early LSA' and the 'Robberg' [46]. Early LSA assemblages show a high degree of variability. Robberg-designated assemblages date to ~18–12 ka and are characterized by systematic bladelet production with rare retouched pieces, that include backed bladelets [47]. Several assemblages across southern African have been designated to the Robberg technocomplex, including Nelson Bay Cave on the south coast, Elands Bay cave on the west coast, Dikbosh 1 at the southern edge of the Kalahari, and Sehonghong in the Lesotho highlands [46].

Mackay et al. [15] hypothesise that MIS 2 was a period of social coalescence, particularly after 22 ka, with wide-spread distribution of the Robberg-designated assemblages across various environments. This is supported by the similarity in flaking systems, implement types, provisioning system, and substantial evidence for different kinds of ornamentation (ostrich eggshell beads, bone beads, and shell pendants). Using stable isotope analysis, Stewart et al. [5] show that OES beads were traded across large distances during MIS 2 into Lesotho, and outside the known range of ostriches (*Struthio camelus*), which indicates long distance macro-scale social networking.

## Ga-Mohana Hill North Rockshelter

Ga-Mohana Hill is located in the southern Kalahari Basin, 12 km northwest of Kuruman on the eastern edge of the Kuruman Hills in South Africa (Fig 1A). The hill has a maximum elevation of 1531 masl, which is about 100-150m above the surrounding landscape. GHN, the largest shelter on the hill, is a long curved and relatively shallow shelter, facing northwest with an impressive view across the landscape, including the Kuruman River. The North of Kuruman Palaeoarchaeology Project began excavating GHN in 2016 and to date has excavated a total of 4.75 m$^2$ of sediment across three areas of the shelter, reaching a maximum depth of 1.7 m [48]. The majority of this (4 m$^2$) has been excavated from Area A, which reveals stratified MSA and LSA deposits (Fig 1B). The top ~10 cm of sediment is loose surface sediment rich in ash and dung, below which are three stratigraphic aggregates. From top to bottom these are Dark Brown Gravelly Silt (DBGS), Orange Ashy Silt (OAS), and Dark Brown Silt and Roofspall (DBSR). Analysis of the slope and orientation of plotted artefacts shows that the artefacts in all these stratigraphic aggregates are in near primary context [48]. Single grain OSL dating on quartz grains has shown that the stratigraphic aggregates date to 14.8±0.8 ka (DBGS), 30.9±1.8 ka (OAS), and 105.2 ±3.3 ka (DBSR) [48,49]. During excavation, all visible artefacts were piece-plotted using a total station.

The focus of this paper is the OES assemblage from the DBGS deposits in Area A dated to ~15 ka. The DBGS contains numerous lithic artefacts, fragmentary faunal remains, charcoal, and rare ochre pieces. The DBGS also includes numerous OES pieces, including fragments, beads, and bead preforms. Overall, the artefact density in the DBGS is 0.42 artefacts per litre of sediment [48]. The lithic artefacts include bladelets and rare backed pieces [48] and based on this are consistent with a LSA, potentially Robberg [47], designation. The lithic artefacts are manufactured primarily from diverse locally available materials including chert and banded ironstone.

## Methods

The study was limited to analysis of OES assemblage from the DBGS stratigraphic aggregate layer in order to investigate and compare MIS 2 sites with OES assemblages across southern

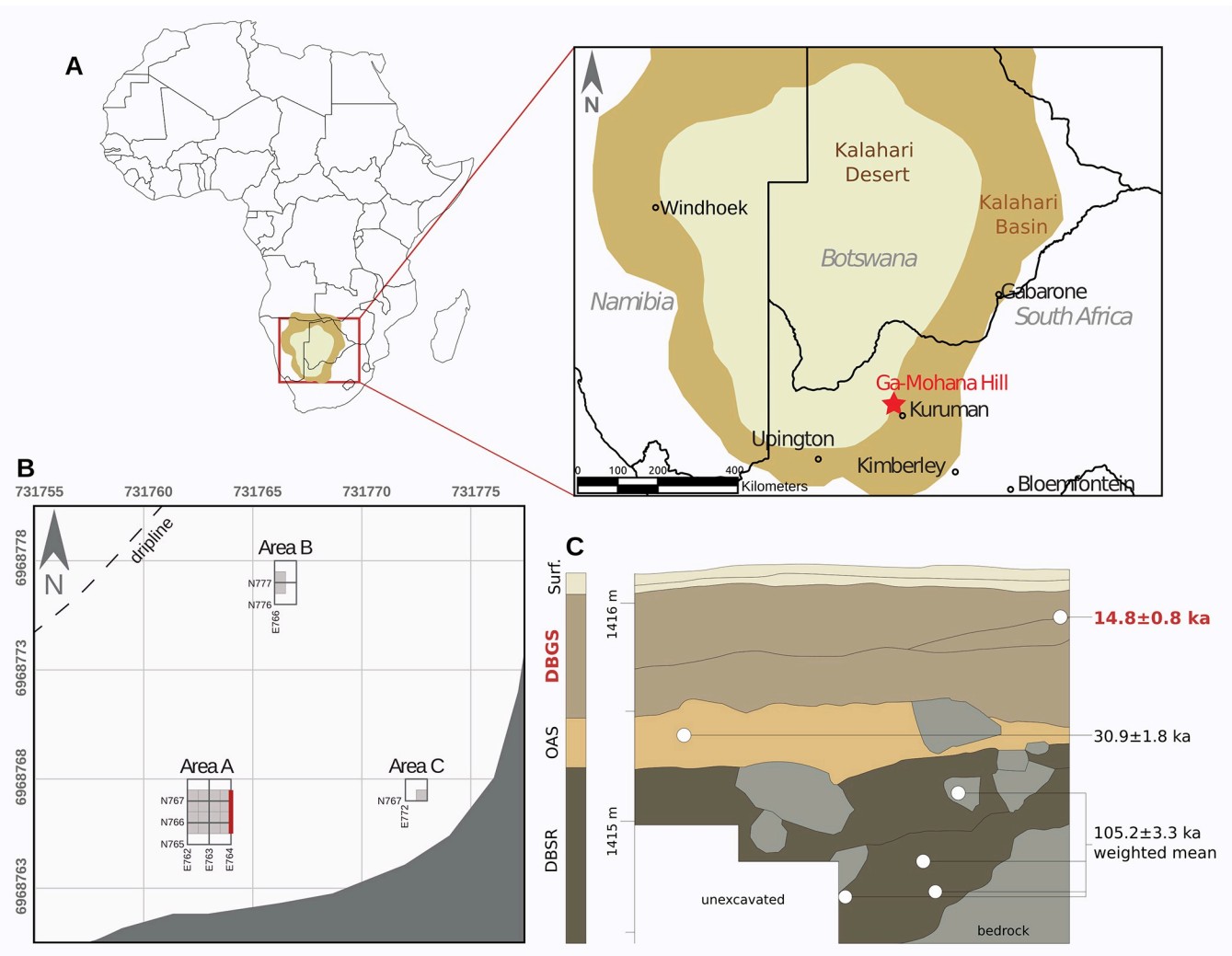

**Fig 1. Map showing the study area and details of Ga-Mohana Hill North Rockshelter.** A—The location of GHN relative to the boundaries of the Kalahari Desert and Basin. B–Map of GHN rockshelter with excavated areas. Red line marks profile shown in C. C -Schematic of stratigraphic boundaries, stratigraphic aggregate assignments, and optically stimulated luminescence sample locations and results. Grey-shaded areas are rock.

Africa. The OES assemblage from layer DBGS at GHN was evaluated following criteria including colour, manufacturing stage, maximum diameter, maximum thickness, aperture diameter, presence of residue, fragment shape [50–54]. The maximum length, width and thickness and fragment shape were recorded for OES fragments.

Colour was attributed to each piece qualitatively, following Collins and Steele [50]. OES colour is a useful tool for understanding whether a shell might have been exposed to heat and the temperature of the heat source. The colouring of beads and fragments ranged from yellow to black (Table 1). OES becomes yellow, red, iridescent and grey when heated under oxidising conditions [32,50,55], while reducing conditions are more likely to produce blackening of the shell. Blackening of OES has not been replicated in experimental studies but is relatively common in the archaeological record [50,52,56]. All beads and fragments were examined with a hand lens (20x magnification) and pieces that showed signs of pigmentation and/or usewear were further examined using a stereo microscope at 10-100x magnification. Pigmentation was classified visually based on colour and texture. Where usewear was identified it was described

**Table 1. Colouring of OES beads, preforms and fragments from the MIS 2 level at GHN.** Colour was assessed following protocol outlined by Collins and Steele [50]. Counts and percentages are recorded.

| Colour | Beads and preforms | | Fragments | |
|---|---|---|---|---|
| | n | % | n | % |
| Unburned | 6 | 21.4 | 7 | 33.3 |
| Yellow | 20 | 71.4 | 10 | 47.6 |
| Red | 0 | 0 | 4 | 19 |
| Black | 2 | 7.2 | 0 | 0 |
| Iridescent | 0 | 0 | 0 | 0 |
| Grey white | 0 | 0 | 0 | 0 |
| Total | 28 | | 21 | |

in terms of location and morphology(facets or striations) following Dayet et al. [51] and Collins et al. [30]. Striations are defined as randomly oriented short marks, while facets are depressions on the surface of the bead. The shape of each fragment was recorded following Miller [10]. Striations, chips, patina and smoothing were recorded for the aperture and outer rims of beads, additionally aperture shape was recorded following Miller [10].

OES beads were assigned a manufacturing stage following Orton's (2008) classification scheme, where stages II-V represent preforms (i.e., beads that have not been completed). Stage II pieces have been partially drilled, but not yet pierced through, while stage III pieces have been completely drilled. Stages IV and V are pieces that have respectively been partially trimmed and completely trimmed. From stage VI pieces are considered finished beads, where stage VI pieces are partially ground and stage VII pieces are completely ground. Finished beads are those that are ready for use as jewellery and/or decoration [30]. Beads were also assigned a manufacturing pathway following Orton [54]. The pathway was determined by presence of drilled but not ground fragments (pathway 1) or ground but undrilled fragments (pathway 2). Beads manufactured following pathway 1 are first drilled and then trimmed, whereas beads made following pathway 2 are first trimmed into round fragments and then drilled. Additionally, it was recorded whether the beads/preforms were broken or complete.

Maximum length and width were measured for all preforms and broken beads that retained less than 50% of their original circumference. Maximum diameter and aperture were measured for finished beads. Maximum thickness was measured for both preforms and beads. While preforms in stages II-V do not have apertures because the drill hole has not completely perforated the shell, the aperture of the drill hole was measured. Usewear traces were noted and identified as facets or striations [30,51].

Fragments were analysed similarly to beads, with maximum length, width and thickness recorded for each fragment. Any marking on the face of the fragments was also noted, both pigment traces and small striations. Any markings were identified as anthropogenic or taphonomic depending on their morphology. Shallow, randomly directed scratches typical of friction from fragments moving in both longitudinal and lateral directions were assigned as taphonomic markings [32]. Intentional markings, characterised as deeper marking with an U or V shaped morphology were designated as anthropogenic. Any other markings that could not confidently be ascribed as taphonomic or anthropogenic were recorded as surface modifications.

Spatial analysis was conducted to examine the patterning of the OES assemblage within the DBGS sediment both vertically and horizontally. All spatial analyses were conducted in R [57], mainly using the *spatstat* package [58]. The spatial patterning of all plotted OES fragments and beads within the excavation area was visually assessed by creating a relative risk surface [59], showing the probability of how likely OES artefacts are to co-occur with other artefact types.

To facilitate comparisons and an understanding of the dynamics of bead manufacture during MIS 2 in southern Africa, a regional comparative database was created. We conducted a literature review to identify well-described and chronometrically dated sites with OES assemblages within southern Africa (South Africa, Lesotho, Eswatini, Namibia, Botswana). We report the level from which the bead assemblages at each site were recovered, as well as the date, where reported, how the beads were classified (as preforms or finished beads), and metric data when available. The classification of finished beads or preforms relies on the initial authors' identification, as much of the data was published prior to the classification schemes used in this analysis [52,54]. These published data were then compared to the GHN results. The maps to illustrate these data were made in the R statistical environment [57]. Data for the ostrich distribution were taken from the South African Bird Atlas Project 2 [60], and then interpolated to cover the entire area (southern Africa). The country border polygons were imported into R from Natural Earth, which provides free vector and raster data using the *rnaturalearth* package [61]. The data on OES beads and fragments from sites across southern Africa were added as a layer to each map to highlight different aspects of the data. This was done using the *ggplot* and *scatterpie* packages [62,63].

The data and code for these maps are available in the supplementary information (S2 Appendix) and in a GitHub repository (https://github.com/amyhatton/ghn_bead_paper). All necessary permits for archaeological investigations at Ga-Mohana Hill were obtained via informed written consent from the South African Heritage Resource Agency (Permit ID 2194). The land is owned by the Baga Motlhware Traditional Council and informed written consent was granted by them to conduct the study. No protected species were sampled and the study did not involve animals. All necessary permits were obtained for the described study, which complied with all relevant regulations. All specimen numbers relevant to this study are provided in S2 and S3 Tables. These specimens are currently housed in the Archaeology Department at the University of Cape Town and they will be permanently curated by the McGregor Museum, Kimberley, Northern Cape, South Africa.

## Results

The OES bead assemblage from layer DBGS consists of 19 beads, 9 bead preforms, and 21 OES fragments. In total 1278 L of sediment was excavated from layer DBGS, thus the density of OES in the sediment is 0.04 pieces per litre of sediment.

### Technological analysis

While the assemblage of OES beads from DBGS represents only 28 beads and preforms, almost all stages of bead manufacture are present [54] (Fig 2, S2 Table). The only stage that is absent is stage III. The presence of preforms and beads in almost all stages of manufacture indicates that OES beads were being manufactured at GHN during MIS 2. The OES beads have a mean aperture diameter of 1.4 mm and a mean external diameter of 4.4 mm (Table 2). Beads and preforms have a mean thickness of 1.6 mm. The external diameter of the OES beads from layer DGBS at GHN falls within the range noted for eastern and southern African foraging populations [13]. More than half of the finished beads (10/19, 53%) have a red residue.

The majority of preforms have been manufactured using Pathway 1, where the bead is first perforated and then trimmed into a disk [54]. One bead may have been manufactured following Pathway 2 where an OES fragment is first trimmed and then perforated to create a bead, however Pathway 2 beads are rare in archaeological assemblages, as they are difficult to distinguish from Pathway 1 beads [54].

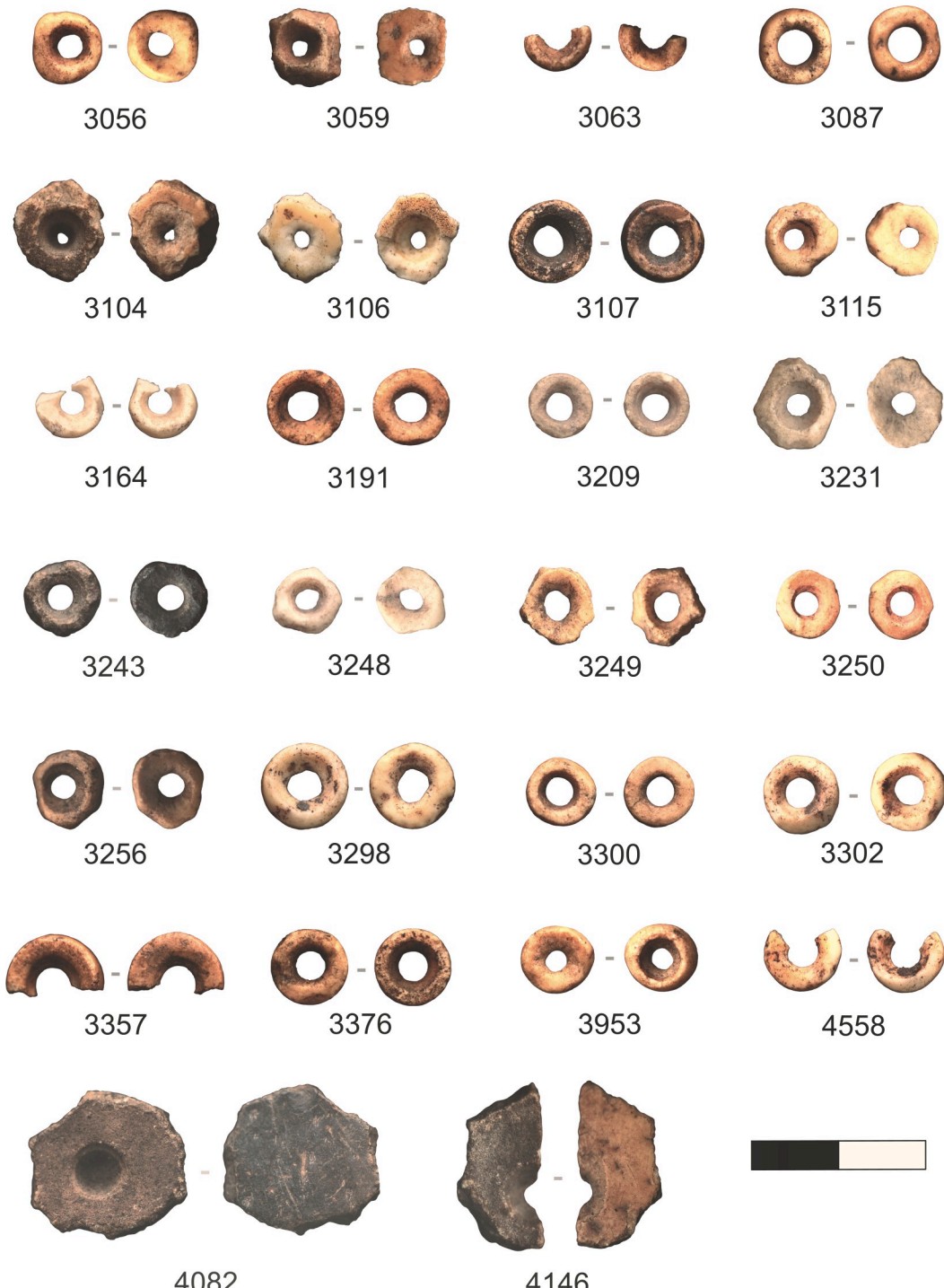

**Fig 2. OES beads and preforms from Layer DBGS at Ga-Mohana Hill North Rockshelter.** Find 4082 was potentially manufactured following Pathway 2. Scale divisions are 5mm.

Twenty-one (21) OES fragments were recovered from the DBGS layer at GHN (S3 Table). The fragments have a mean length of 14.6 mm (range = 9.69 to 21.95, sd = 2.9), a mean width of 10 mm (range = 4.58 to 14.73, sd = 2.6) and a mean weight of 0.49 g (range = 0.12 to 0.84,

**Table 2. Technological features for the OES beads and preforms from the MIS 2 deposit at Ga-Mohana Hill North Rockshelter.** Complete beads are those identified as stage VI and VII following Orton [54]. Standard deviations are included in brackets where applicable.

| Technological attributes | Finished beads | Preforms |
|---|---|---|
| # unbroken beads | 16 | 1 |
| # broken beads | 3 | 8 |
| total # beads | 19 | 9 |
| mean exterior diameter in mm (standard deviation) | 4.4 (0.42) | - |
| exterior diameter range in mm | 3.69–5.45 | - |
| mean aperture diameter in mm (standard deviation) | 1.5 (0.25) | - |
| aperture range in mm | 1.04–2.05 | - |
| mean thickness in mm (standard deviation) | 1.6 (0.19) | |
| thickness range in mm | 1.2–1.8 | |
| # complete beads with use-wear | 13 | - |
| # of beads and preforms with residue | 10 | |
| # of completed beads with both use-wear and residue | 7 | - |

sd = 0.2). The only two shapes represented in the fragments are polygonal and triangular. Most of the pieces were a polygon shape (n = 15), with the remaining six being a triangular shape. The OES fragments from the DBGS layer at GHN amount to 10.3 g, which represents a minimum of 1 ostrich eggshell. Commercial ostrich eggshells average at 222 g, with a range of 180–292 g [64]. Five of the 21 fragments (24%) exhibited red residue staining on their surface.

## Taphonomy

Many fragments had evidence of scratches on their surface. These randomly directed shallow marks are likely of taphonomic origin from OES pressing against harder objects in the sediment, possibly through trampling [32]. Both beads and fragments have similarities in level of heat exposure for layer DBGS at GHN, with the majority in the yellow category which is indicative of OES being heated to about 200 ˚C [50]. There are however a few differences between the burning patterns of beads and preforms compared to fragments, particularly for those classified as black or red. No beads were classified as red, however 19% of the OES fragment assemblage is red, which indicates that these fragments were heated to between 300–350 ˚C. About 7% of the beads and preforms are blackened, showing that they were likely burned in an environment with limited oxygen, whereas no fragments were blackened. These differences are statistically significant, however the sample size is small (chi-squared test, $\chi2 = 8.59$, $p = 0.035$).

## Spatial analysis

The artefacts are uniformly spaced horizontally across excavation Area A at GHN except for the SE corner (Fig 3) where a large block of roofspall was removed during excavations. There is slight patterning in the distribution of OES; OES artefacts are most likely to occur with non-OES artefacts in the northeast corner of the excavation (Fig 3B). Vertically, artefacts are quite evenly spaced through the DBGS, but most OES artefacts occur in the upper 40 cm of the stratigraphic aggregate (Fig 3). There is one OES fragment that occurs lower than other OES artefacts, in the southeast corner. The relative risk surface shows that laterally there are two clusters where OES artefacts are likely to occur with non-OES artefacts (Fig 3A and 3C).

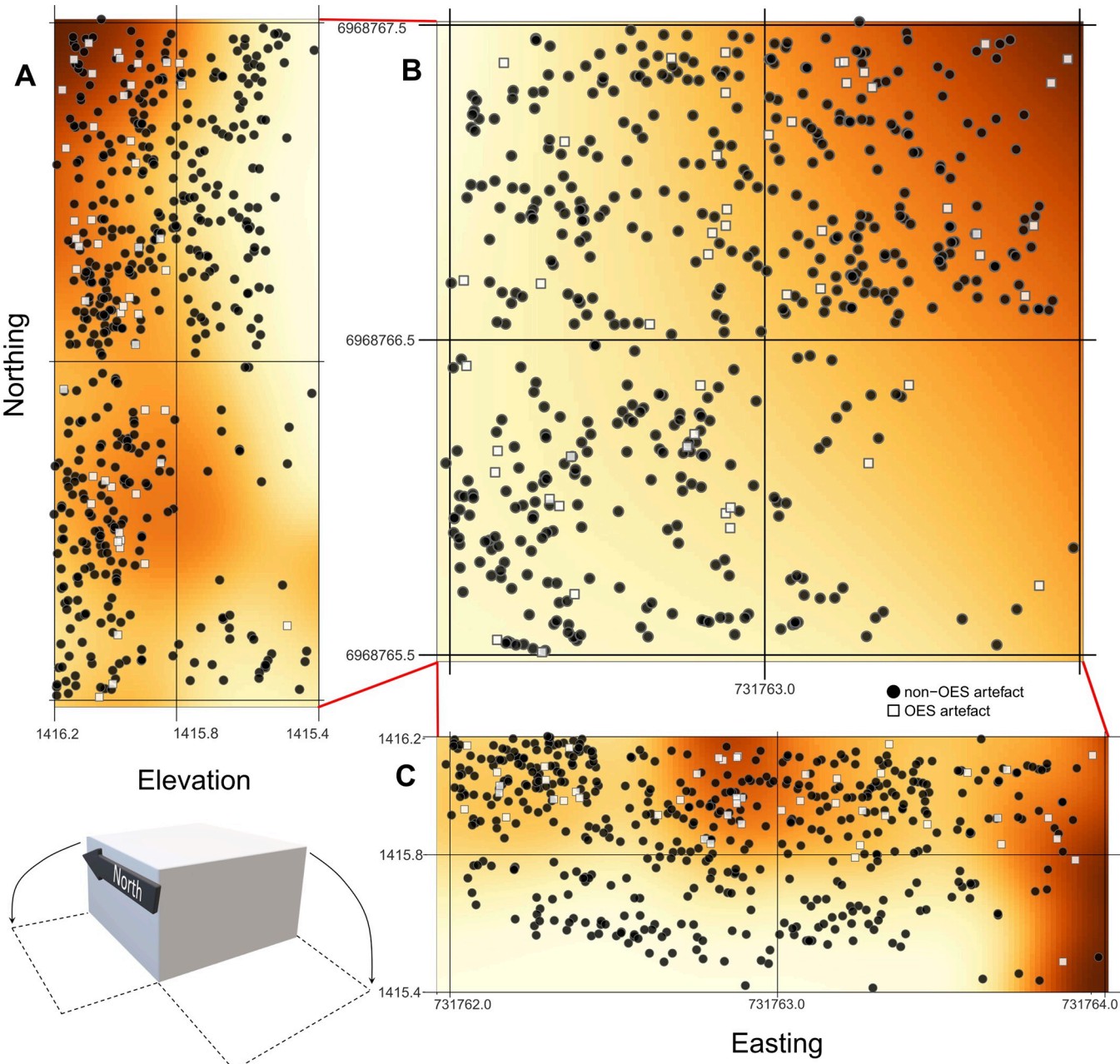

**Fig 3. Relative risk surface for OES artefacts (fragments and beads) compared to all other plotted artefacts in DBGS layer at GHN.** Non-OES artefacts are shown as black circles while OES artefacts are shown as white squares. The darker orange indicates areas where OES artefacts and other artefacts have a higher probability of co-occurring, while lighter areas are indicative of areas where OES artefacts and other artefacts are not likely to co-occur. A- lateral view of DBGS layer looking west. B- Aerial view of DBGS layer. C- lateral view of DBGS looking north.

## Regional comparison

We reviewed the literature to synthesise data on OES bead assemblages from MIS 2 (Tables 3 and S1) and situate the OES bead assemblage from GHN in regional context. OES beads and fragments were unevenly reported across sites, which makes it challenging to compare quantitatively. However, a few useful observations and comparisons can be made. The majority of assemblages are small, with fewer than 50 OES beads and preforms (Fig 6). The exceptions are

**Table 3. MIS 2 dated sites in southern Africa with OES assemblages.** Modern ostrich prevalence has been calculated from SABAP 2 [60].Ostrich sighting data was interpolated across southern Africa and the values included here are the mean of a 5km radius around each site location. More detailed information on the sites and OES assemblages are available in S1 Table.

| Site | Age | OES assemblage description | Modern ostrich prevalence (sighting percentage) | References |
|---|---|---|---:|---|
| Sehonghong | 14.5–25 ka | 9 finished beads and no preforms or fragments. | 17 | [65,66] |
| Ha Makotoko | 15–29ka | 12 finished beads and no preforms or fragments. Mean diameter of beads is 3.1mm. | 30 | [67,68] |
| Melkhoutboom Shelter | 18.7 ka | 5 finished beads and 6 preforms, there were 393 fragments of which 2 were decorated. | 37 | [69] |
| Apollo 11 | 13–14.5 ka | 4481 fragments, but no beads or preforms. | 44 | [70] |
| Nelson Bay Cave | 11.7–14.5 ka | 80 finished beads and 53 preforms. 35 fragments were reported. | 13 | [71–73] |
| Bushman's Rock Shelter | 10–13 ka | 83 finished beads, 101 preforms and 419 fragments. The mean bead diameter is 5.3mm. | 19 | [31] |
| Umhlatuzana | 16 ka | 4 finished beads, no preforms or fragments. | 21 | [74,75] |
| Rose Cottage Cave | 16 ka | 1 finished bead and 25 fragments. | 36 | [76,77] |
| Boomplaas | 15–22 ka | 63 finished beads, 13 preforms, 4025 fragments. | 26 | [71,78,79] |
| Ntloana Tsoana | 14 ka | 2 finished beads. | 30 | [68] |
| Spitzkloof A | 23.5 ka | 2 finished beads and 2 preforms. 1179 OES fragments. | 34 | [80] |
| Ga-Mohana Hill North Rockshelter | 15 ka | 19 finished beads, 9 preforms and 21 fragments. Mean bead diameter is 4.4mm. | 48 | [48] |
| Drotsky's Cave (Gcwihaba Cave) | 14.5 ka | 2 finished beads and 197 fragments. | 46 | [81] |
| Dikbosch 1 | 14.5–16.5 ka | 48 finished beads, 44 preforms and 12157 fragments. Mean bead diameter is 4.5mm. | 69 | [16,82] |
| Heuningneskrans | 23 ka | 1 finished bead. | 21 | [83] |
| Buffelskloof | 27 ka | 10 finished beads, 30 preforms. Mean bead diameter is 4.3mm. | 25 | [84] |
| Grassridge | 11.6–13.5 ka | 9 finished beads, 28 preforms and 573 fragments. Mean bead diameter is 3.5mm. | 35 | [30,85,86] |
| Txina Txina | 25–29 ka | 2 finished beads and 28 fragments. | 17 | [87] |

Nelson Bay Cave, Boomplaas, Bushman's Rockshelter and Dikbosch, which date to the terminal part of MIS 2 and where bead counts are in the range of 74 to 170. Interestingly, all sites within Lesotho/Maloti Drakensberg have very low counts of beads and preforms compared to sites in the rest of southern Africa (Fig 5).

Sites in areas where modern ostrich prevalence is low have high proportions of finished beads to preforms (Fig 4). The exception is Heuningneskrans which has no preforms but is in an area with moderate ostrich prevalence. Interestingly, at BRS which is 38 km away, there is a roughly equal proportion of preforms to finished beads. The Heuningneskrans assemblage does however only consist of 1 OES bead currently.

Ntloana Tsoana, Ha Makotoko, Sehnghong, Umhlatuzana, and Heuningneskrans are the only sites with high proportions of finished beads compared to preforms and fragments of OES, most other sites are dominated by high proportions of preforms and fragments (Fig 5). The number of recorded OES fragments for sites across southern Africa ranges from 1 to 12157 (Fig 7), although most sites have less than 1000 fragments recorded.

We compared mean bead diameter for occupations from sites that recorded this information (Fig 8) and found that the mean bead diameters are significantly different to one another

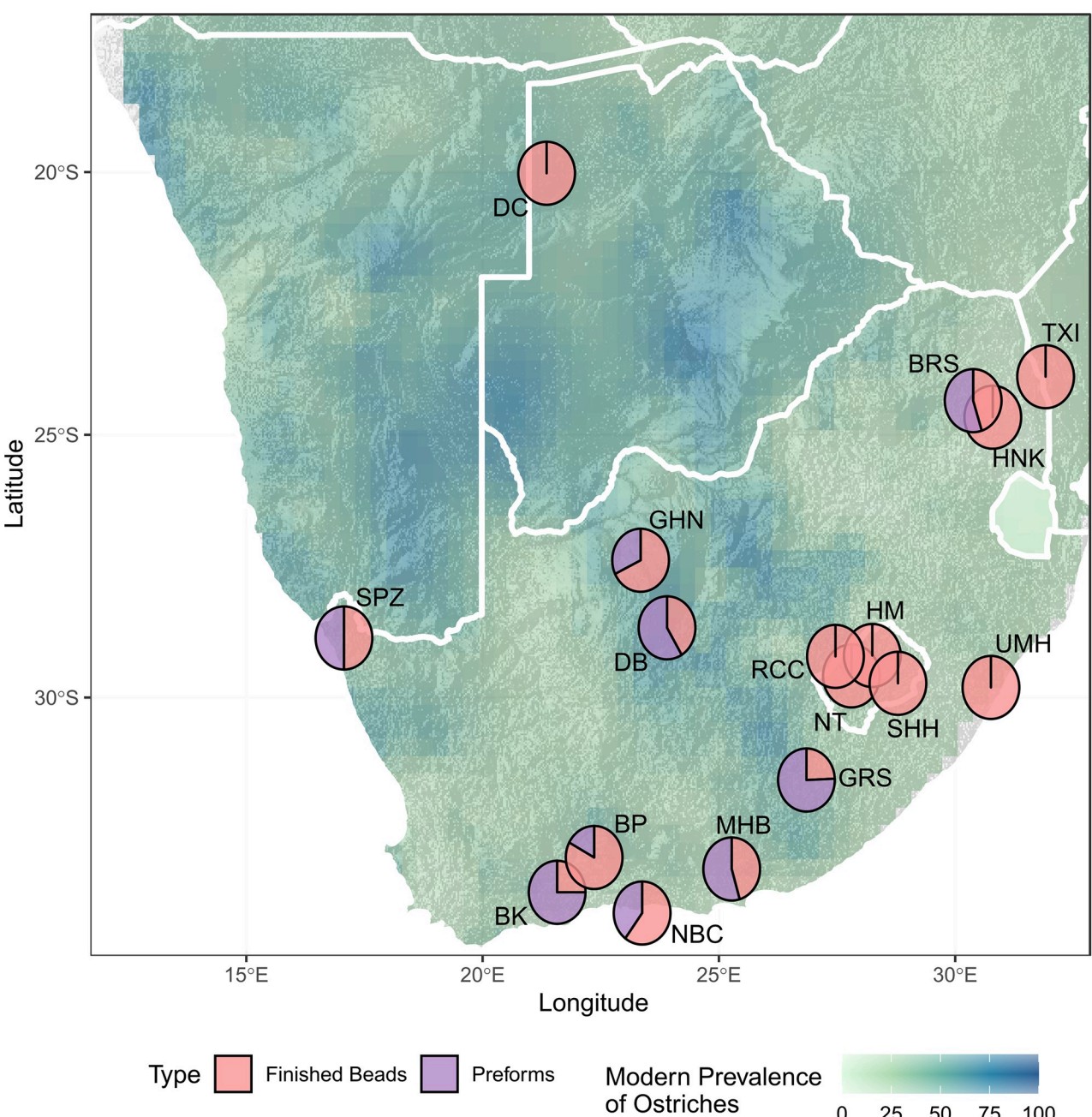

**Fig 4. Map of southern Africa showing archaeological sites dated to MIS 2 that have OES bead assemblages.** Pie charts show the proportion of preforms to finished beads at each site, overlaid on a map of modern ostrich prevalence based on South African Bird Atlas Project 2 data [60]. SPZ–Spitzkloof A, BP–Boomplaas, BK–Buffelskloof, NBC–Nelson Bay Cave, MHB–Melkhoutboom, GRS—Grassridge Rockshelter, NT–Ntloana Tsoana, RCC–Rose Cottage Cave, SHH–Sehonghong, HM–Ha Makotoko, UMH–Umhlatuzana, BRS–Bushmans Rockshelter, HNK—Heuningneskrans, TXI —Txina Txina, GHN–Ga-Mohana Hill North, DB–Dikbosch, DC—Drotsky's Cave (Gcwihaba Cave).

(ANOVA, F = 9.935, $p$ = 0.0124). A Tukey pairwise comparison shows that mean bead diameters differ significantly between Grassridge and Bushman's Rock Shelter ($p$ = 0.045), Ha-Makotoko and Bushman's Rock Shelter ($p$ = 0.008), and finally Ha-Makotoko and Dikbosch ($p$ = 0.03).

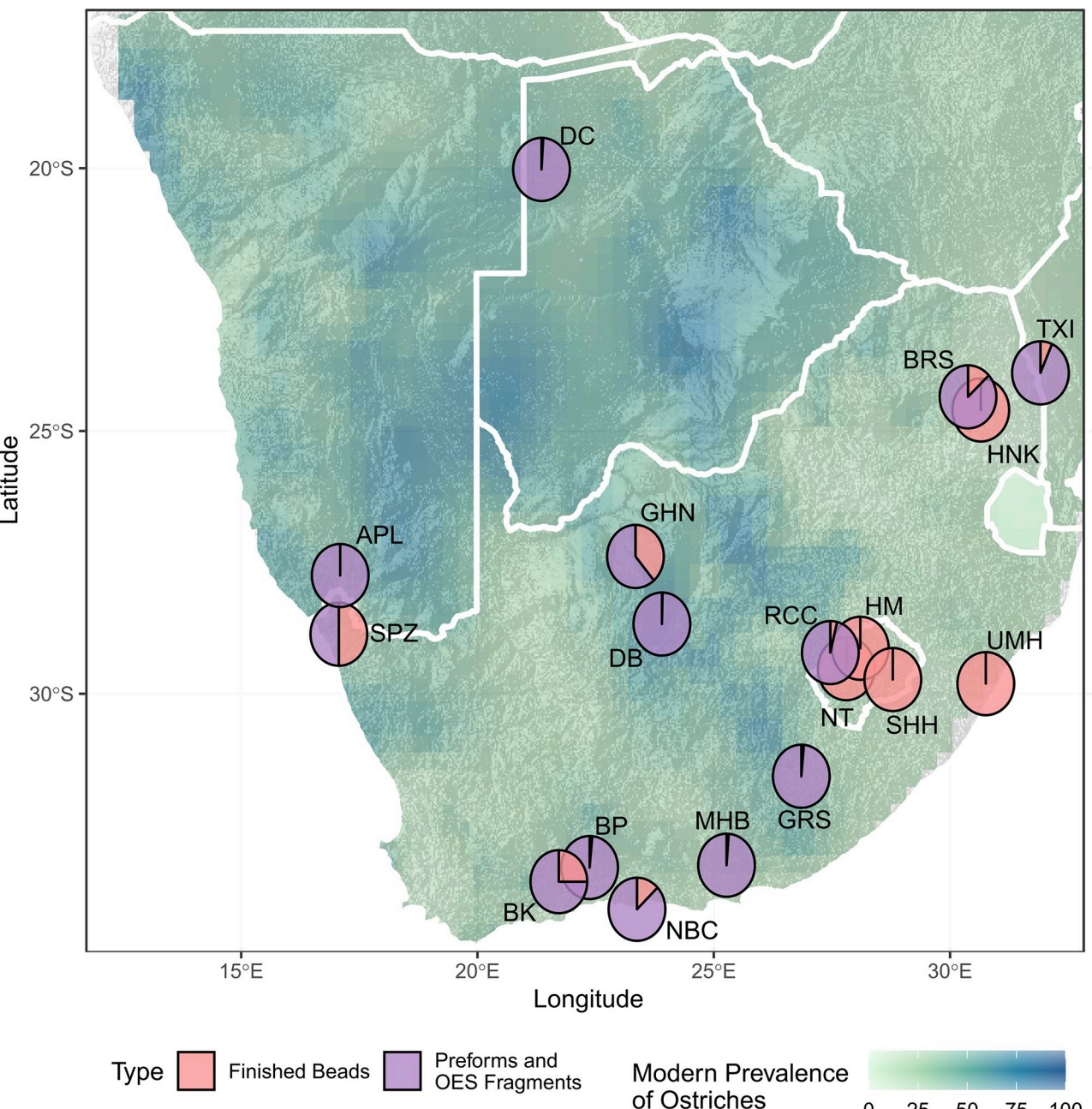

**Fig 5. Map of southern Africa showing archaeological sites dated to MIS 2 that have OES bead assemblages.** Pie charts show the proportion of finished beads to preforms and OES fragments at each site, overlaid on a map of modern ostrich prevalence based on South African Bird Atlas Project 2 data [60]. BK–Buffelskloof, BP–Boomplaas, NBC–Nelson Bay Cave, MHB- Melkhoutboom, GRS—Grassridge Rockshelter, NT–Ntloana Tsoana, RCC– Rose Cottage Cave, HM—Ha Makotoko, SHH–Sehonghong, UMH–Umhlatuzana, BRS–Bushmans Rockshelter, HNK—Heuningneskrans, TXI— Txina Txina, GHN–Ga-Mohana Hill North, DB—Dikbosch, DC—Drotsky's Cave (Gcwihaba Cave), APL—Apollo 11, SPZ—Spitzkloof.

## Discussion and conclusions

### Bead manufacture

The presence of OES beads in all stages of manufacture suggests that beads were being made at GHN ~15 ka. There is however a higher proportion of beads compared to preforms, which

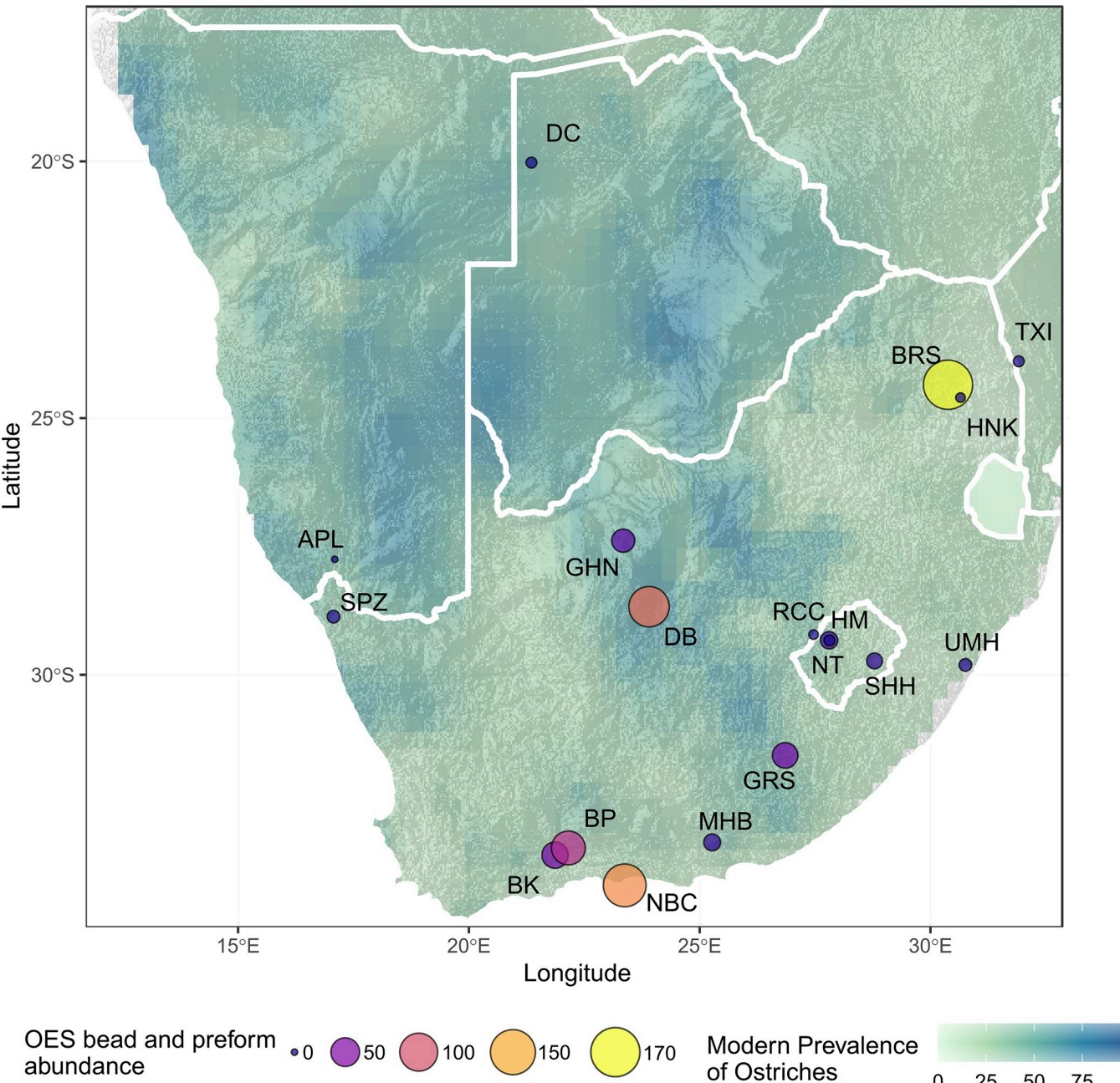

**Fig 6. Map of southern Africa showing archaeological sites dated to MIS 2 that have OES bead assemblages.** OES bead counts for sites, overlaid on a map of modern ostrich prevalence based on South African Bird Atlas Project 2 data [60]. BK–Buffelskloof, BP–Boomplaas, NBC–Nelson Bay Cave, MHB-Melkhoutboom, GRS—Grassridge Rockshelter, NT–Ntloana Tsoana, RCC–Rose Cottage Cave, HM—Ha Makotoko, SHH–Sehonghong, UMH–Umhlatuzana, BRS–Bushmans Rockshelter, HNK—Heuningneskrans, TXI—Txina Txina, GHN–Ga-Mohana Hill North, DB—Dikbosch, DC—Drotsky's Cave (Gcwihaba Cave), APL—Apollo 11, SPZ—Spitzkloof.

could indicate that drilling and grinding of beads were conducted in different areas of the rockshelter. OES fragments are outnumbered by worked OES, which is interesting and may also suggest that bead manufacture took place in an as yet unexcavated part of the rockshelter, or perhaps that OES was used in a conservative manner during this period at GHN. Extending the spatial extent of this excavation will provide more clarity.

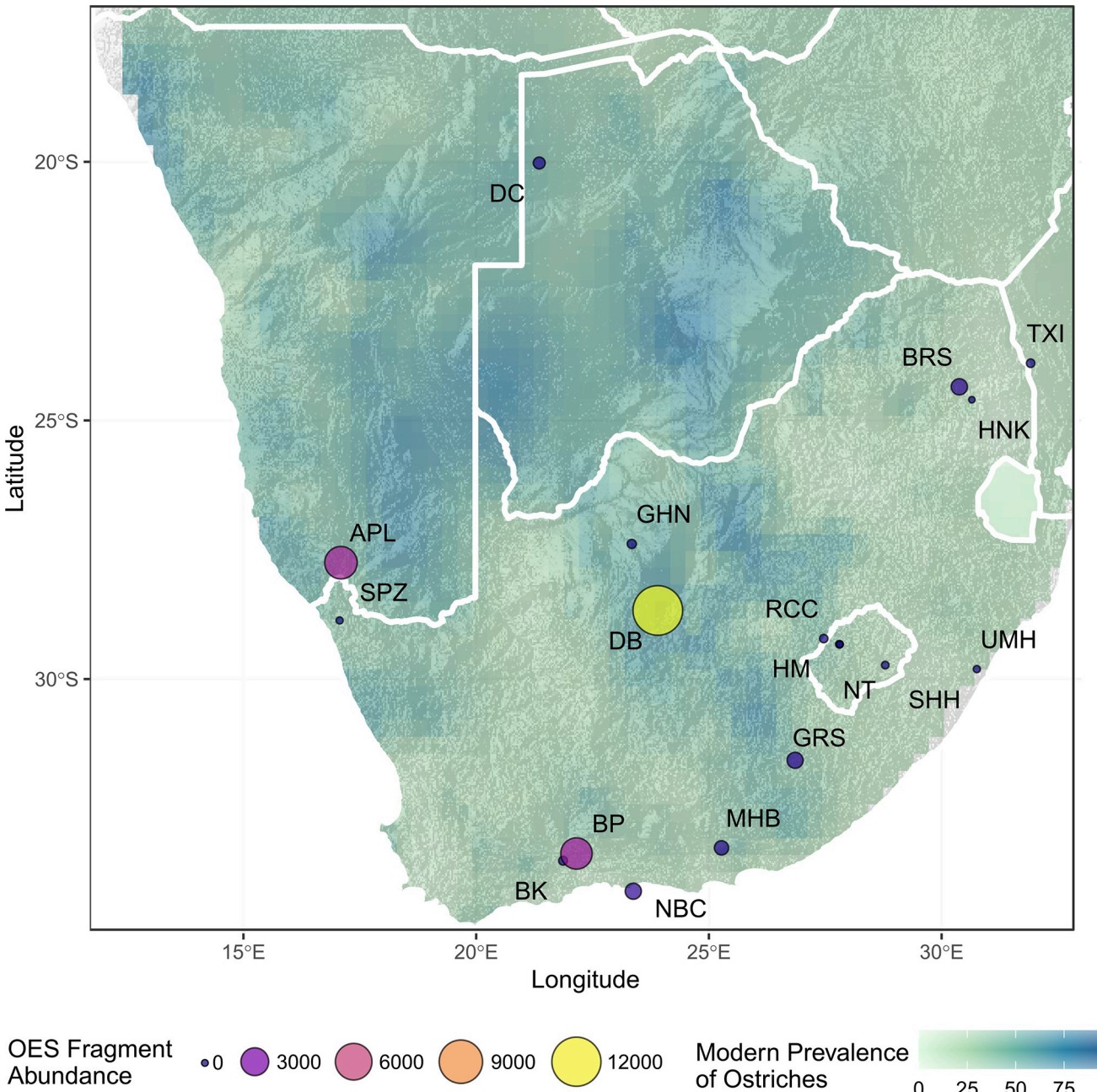

**Fig 7. Map of southern Africa showing archaeological sites dated to MIS 2 that have OES bead assemblages.** OES fragment counts for sites, overlaid on a map of modern ostrich prevalence based on South African Bird Atlas Project 2 data [60]. BK–Buffelskloof, BP–Boomplaas, NBC–Nelson Bay Cave, MHB-Melkhoutboom, GRS—Grassridge Rockshelter, NT–Ntloana Tsoana, RCC–Rose Cottage Cave, HM—Ha Makotoko, SHH–Sehonghong, UMH–Umhlatuzana, BRS–Bushmans Rockshelter, HNK—Heuningneskrans, TXI—Txina Txina, GHN–Ga-Mohana Hill North, DB—Dikbosch, DC—Drotsky's Cave (Gcwihaba Cave), APL—Apollo 11, SPZ—Spitzkloof.

It is interesting to note that modern ostrich prevalence at GHN is quite high (48%, Table 3). Although data on changes in ostrich distribution in the late Quaternary are limited, if we assume that ostrich distribution has been relatively consistent across wet and dry phases over the last 15 ka [88], then ostrich eggs and their shells would have been regularly encountered. The relatively small number of both recovered OES beads and recovered OES fragments,

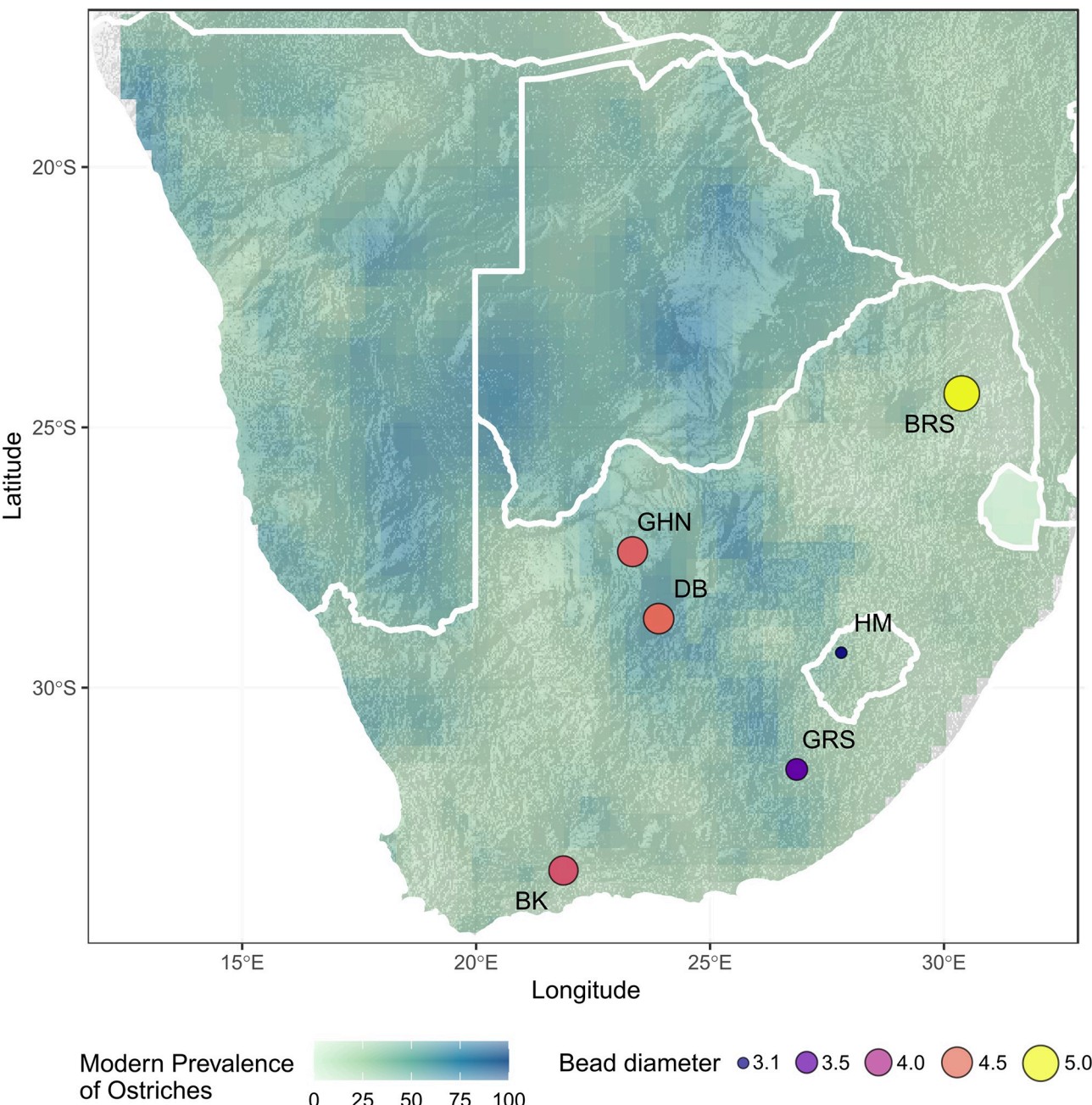

**Fig 8. Map of southern Africa showing archaeological sites dated to MIS 2 that have OES bead assemblages.** Mean OES bead diameters for sites, overlaid on a map of modern ostrich prevalence based on South African Bird Atlas Project 2 data [60]. BK–Buffelskloof, GHN–Ga-Mohana Hill North, DB—Dikbosch, GRS—Grassridge Rockshelter, HM-Ha Makotoko, BRS–Bushmans Rockshelter.

however, suggest that these activities may not have been intensively practiced at GHN during this occupation and may reflect less intensive site use. Further excavation expanding the spatial extent of this occupation at GHN will further inform this hypothesis.

Of interest is the bead that was potentially manufactured using Pathway 2, as well as the black beads. Beads manufactured using Pathway 2 are rare in the archaeological record, with Orton [54] noting 5 beads conforming to Pathway 2 out of 465 from the late Holocene site

KN2006/067 in Namaqualand, South Africa. One reason for this rarity, is that this manufacturing pathway is challenging to identify unless recovered in Stages II-IV. Once the hole has punctured through the OES preform it is difficult to know whether the bead was trimmed or drilled first. This bead at GHN is currently the oldest described bead that may have been manufactured following Pathway 2 in southern Africa and may suggest that diverse strategies for the manufacture of OES beads are an early characteristic of this technology. Further consideration and research are needed to better understand how differences in manufacturing strategies relate to resource availability, social context of production, and potentially inform cultural inheritance.

One bead and one preform (Stage II) recovered from DBGS were coloured black. This finding is of interest because of the absence of black colouration in the OES fragments, as well as the specific conditions noted for obtaining a black colour for OES in general [50,52,56]. The absence of OES fragments that were blackened suggests that this may reflect a deliberate behaviour for the colouration of beads, in that we would also expect to see black OES fragments if the colouration resulted from taphonomic processes. However, the data on the colouration of OES fragment assemblages is largely unreported, which limits comparisons with other sites. Further taphonomic research, especially regarding OES fragment assemblages, is required to better link bead colour to anthropogenic intent.

There is some evidence of red coloured residues located within the aperture and/or associated with wear facets of OES beads, which may indicate that the bead came into contact with an ochred surface prior to deposition. However, the presence of red coloured residues on fragments and randomly positioned on beads suggests that these residues likely result from contact with ochre within the sediment peri- or post-deposition [30].

## Spatial patterning

Ethnographic data on OES bead manufacture is limited, which in turn limits our understanding of how beads were manufactured. Where OES bead manufacturing activities take place within sites is also poorly understood. It is possible that such social activities might have taken place near to hearths, however this needs to be determined by more research into spatial patterning of OES. At GHN, distinct hearth features are not identifiable in the sediments of Area A, despite evidence for burning (burned bone, charcoal). While there is some spatial patterning, in general, the distribution of OES at GHN is not strongly clustered in any one specific area.

Many of the OES artefacts are found in small clusters of 2–5 artefacts that could relate to the manufacturing process. Most of these clusters are made up of OES fragments, so it is possible that these fragments broke once they had been deposited in the sediment.

Extending the excavation area at GHN will allow for a more nuanced understanding of OES manufacture at the site and shed light on whether there was spatial partitioning of intensive OES bead manufacture in the rockshelter.

## Regional context

Our literature review identified several sites that contain OES bead assemblages dating to the latter part of MIS 2 in southern Africa (Tables 3 and S1). This review identifies the presence of OES beads across much of southern Africa during this period and emphasises their importance for terminal Pleistocene foragers. OES beads reflect decorative objects and the practice of culturing the body [9], and therefore their presence suggests that this practice was embodied and persistent across the foraging groups in southern Africa during this time. This conclusion is consistent with OES bead research focusing on Holocene assemblages [13].

Moreover, this conclusion is also consistent with the argument for MIS 2 being a period of social coalescence within southern Africa [15]. Importantly, the terminal Pleistocene effectively demonstrates the first period in the southern African archaeological record where we see one type of non-lithic stylistic object occur across southern Africa. As stated above, the presence of OES beads across southern Africa indicates their cultural and social importance and further emphasises that the importance and meaning of OES bead decorative objects was transmitted and shared between populations and groups living in diverse parts of the sub-continent. In this regard, the presence of OES beads across southern Africa during the terminal Pleistocene indicates region-wide social connections, and aligns with the lithic data presented by Mackay et al. [15] for a period of social coalescence during MIS 2. The persistence, and indeed increase, in OES bead assemblages (both in terms of numbers of beads, as well as sites demonstrating beads) [10,13], speaks to the strengthening of these social connections during MIS 1.

However, there are stylistic differences in terms of bead diameter across southern Africa during the terminal Pleistocene (Fig 8). Of note are the significant differences in bead diameter between Grassridge and Bushman's Rock Shelter, Ha-Makotoko and Bushman's Rock Shelter, and Ha-Makotoko and Dikbosch 1. Ha-Makotoko and Bushman's Rock Shelter have the smallest and largest bead diameters respectively, and reflect a diversity in mean bead diameter across southern Africa during late MIS 2 that ranges from 3 to more than 5 mm. These differences may reflect local stylistic variation and preferences, potentially with large-sized beads being favoured in the northeast of southern Africa, as indicated by Bushman's Rock Shelter [31,51], intermediate-sized beads in the western part of southern Africa, as indicated by Buffelskloof, Dikbosch1, and GHN, and small-sized in the Drakensberg and sub-escarpment as indicated by Ha-Makotoko and to a lesser extent Grassridge Rockshelter (Fig 8). This bead size variation does not appear to correlate with the prevalence of ostriches (Spearman's Rank Correlation, rho = -0.02, p-value = 1).

The absence of preforms suggests that beads were likely not manufactured at Ha-Makotoko, where bead diameters are the smallest and therefore, the majority of OES beads are argued to have been imported to the region [4,5]. The movement of OES beads over potentially hundreds of kilometres complicates our understanding of regional bead diameter diversity. Use-wear and taphonomic processes may also affect bead diameter [89]. However, the nature of these processes is not yet well understood, and the OES beads from Ha-Makotoko have yet to undergo a taphonomic analysis. That being said, the OES beads from Ha-Makotoko cluster around a mean diameter of 3.1 mm, with a tight range from 3–3.2mm, and are suggestive of a preference or requirement for this smaller bead size.

These data therefore indicate potential pockets of localised stylistic variation in OES bead manufacture. This is of interest, as MIS 2 is suggested to be a period of social coalescence, in part indicated by the widespread occurrence of sites attributed to the Robberg technocomplex. However, the use of a lithic technocomplex likely masks diversity in lithic assemblage compositions during this period, and that technological variation becomes much stronger after 14 ka [15]. This pattern seems to fit with the OES bead data discussed above, in terms of increasing local influences and stylistic diversity, and suggests that while social networks were persistent during this period, there was also an increase in local stylistic innovation. The increasing diversity in material culture during this period may potentially relate to the diverse environmental responses to global cooling across southern Africa during MIS 2 [42], but more work focused on local environmental contexts at these various sites is required.

From a local perspective, the site nearest to GHN is Dikbosch 1 [16,82], located 152 km from GHN. Dikbosch 1 has four occupation layers that date to MIS 2, these are dated as contemporary and slightly earlier than GHN with a range of 14.5–16.6ka (Figs 4–8, S1 Table).

OES bead sizes are similar in terms of mean diameter, which is suggestive of stylistic, and potentially social, continuity between the two sites. However, the sites differ in terms of number of beads, ratio of beads to preforms, and size of the OES fragment assemblages (Figs 4, 6 and 7; Table 3). Dikbosch 1 has a larger OES bead assemblage both in terms of number of OES beads and preforms. The ratio of beads to preforms also differs, with Dikbosch 1 demonstrating more preforms to beads, as opposed to GHN, where we see more beads than performs. Moreover, Dikbosch 1 has a much larger OES fragment assemblage. Part of this difference may relate to Dikbosch 1 being located in an area with a higher prevalence of ostriches at 69% compared to the 48% at GHN (Table 3). The scale of the difference, especially with regard to the OES fragment assemblages, also suggests that ostrich eggs and OES may have been of more importance, both as a subsistence resource, as well as a raw material, at Dikbosch 1 and that bead manufacture was likely more intensive at this site. In this respect, we suggest that OES beads were being manufactured at GHN, but that the practice was not as intensive as other sites that have been described as "bead factories" [30,54].

## OES bead reporting

Also of note is the lack of standardised reporting for OES bead assemblages in the literature. This in part reflects the historical lack of attention given to OES assemblages (both beads and fragments) in archaeological research, perhaps because they were considered to be of less interpretive value than other artefacts classes. Regardless, presenting detailed descriptions of OES beads (including major attribute data), as well as OES fragments, provides important insight into past behaviours at local, regional, and sub-continental scales, as well as taphonomic information that informs site formation processes. Many sites we discuss in this review lack much of these data restricting our understanding of the role of OES and OES beads in the past. In this respect, we concur with Miller and Wang [14], Miller [10], Miller and Sawchuk [13], Collins et al. [30], and Collins [9] in arguing for greater attention to OES and OES bead assemblages, and specifically for providing crucial (and fundamental) technological data for these assemblages. We now know the practice of culturing bodies with OES beads was widespread geographically across southern and eastern Africa and China during the Late Pleistocene. Studying the manufacture and distribution of beads in detail can offer further opportunity to understand social interaction between people on the landscape during MIS 2.

## Supporting information

**S1 Table. Southern African MIS 2 ostrich eggshell assemblages.** Database of ostrich eggshell assemblages at archaeological sites dating to between 29 and 12ka.
(CSV)

**S2 Table. Ga-Mohana Hill North Rockshelter ostrich eggshell bead data.** Technological data for each plotted find ostrich eggshell bead from the DBGS level at Ga-Mohana Hill North Rockshelter.
(CSV)

**S3 Table. Ga-Mohana Hill North Rockshelter ostrich eggshell fragment data.** Technological data for each plotted find ostrich eggshell fragment from the DBGS level at Ga-Mohana Hill North Rockshelter.
(CSV)

**S1 Appendix. Code for making the maps of southern African MIS 2 ostrich eggshell assemblages.** PDF document of RMarkdown version of the code for making the maps of southern

African MIS 2 ostrich eggshell assemblages.
(PDF)

**S2 Appendix. Code for statistical analysis of ostrich eggshell data.** PDF document of RMarkdown version of the code for statistical analysis of southern African MIS 2 ostrich eggshell assemblages and Ga-Mohana Hill North Rockshelter assemblages.
(PDF)

## Acknowledgments

We pay respects to the traditional owners of the land on which GHN is located and thank the Baga Motlhware Traditional Council and South African Heritage Resources Agency for permission to work at Ga-Mohana Hill. Fieldwork at GHN is part of the North of Kuruman Palaeoarchaeology project and we would like to thank our collaborators on that project, and in particular Kyle Brown, Robyn Pickering, Luke Glliganic, and Sechaba Maape for their contributions to the archaeology and geochronology of the site. We would also like to acknowledge all those who have participated in excavation, including many student volunteers. Thank you also to David Morris and the McGregor Museum, and to the staff and faculty at the University of Cape Town.

## Author Contributions

**Conceptualization:** Amy Hatton, Benjamin Collins, Benjamin J. Schoville, Jayne Wilkins.

**Data curation:** Amy Hatton, Benjamin Collins, Benjamin J. Schoville, Jayne Wilkins.

**Formal analysis:** Amy Hatton, Benjamin Collins, Benjamin J. Schoville.

**Funding acquisition:** Benjamin J. Schoville, Jayne Wilkins.

**Investigation:** Amy Hatton, Benjamin Collins, Benjamin J. Schoville, Jayne Wilkins.

**Methodology:** Amy Hatton, Benjamin Collins, Benjamin J. Schoville, Jayne Wilkins.

**Resources:** Amy Hatton, Benjamin Collins, Benjamin J. Schoville, Jayne Wilkins.

**Visualization:** Amy Hatton.

**Writing – original draft:** Amy Hatton, Benjamin Collins, Benjamin J. Schoville, Jayne Wilkins.

**Writing – review & editing:** Amy Hatton, Benjamin Collins, Benjamin J. Schoville, Jayne Wilkins.

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
