## [Decision Letter · Decision Letter 0]

3 Jan 2022

PONE-D-21-22127Ostrich eggshell beads from Ga-Mohana Hill North Rockshelter, southern Kalahari, and the implications for understanding social networks during Marine Isotope Stage 2.PLOS ONE

Dear Dr. HATTON,

Thank you for submitting your manuscript to PLOS ONE. After careful consideration, we feel that it has merit but does not fully meet PLOS ONE’s publication criteria as it currently stands. Therefore, we invite you to submit a revised version of the manuscript that addresses the points raised during the review process.

We look forward to receiving your revised manuscript.

Kind regards,

Enza Elena Spinapolice, Ph.D

Academic Editor

PLOS ONE

Journal Requirements:

2. In your manuscript, please provide additional information regarding the specimens used in your study. Ensure that you have reported specimen numbers and complete repository information, including museum name and geographic location.

For more information on PLOS ONE's requirements for paleontology and archaeology research, see https://journals.plos.org/plosone/s/submission-guidelines#loc-paleontology-and-archaeology-research.

4. We note that Figures 3, 4, 5, 6, and 7 in your submission contain map images which may be copyrighted. All PLOS content is published under the Creative Commons Attribution License (CC BY 4.0), which means that the manuscript, images, and Supporting Information files will be freely available online, and any third party is permitted to access, download, copy, distribute, and use these materials in any way, even commercially, with proper attribution. For these reasons, we cannot publish previously copyrighted maps or satellite images created using proprietary data, such as Google software (Google Maps, Street View, and Earth). For more information, see our copyright guidelines: http://journals.plos.org/plosone/s/licenses-and-copyright.

a. You may seek permission from the original copyright holder of Figures 3, 4, 5, 6, and 7 to publish the content specifically under the CC BY 4.0 license.  

Reviewers' comments:

Reviewer's Responses to Questions

**Comments to the Author**

1. Is the manuscript technically sound, and do the data support the conclusions?

Reviewer #1: Yes

Reviewer #2: Yes

2. Has the statistical analysis been performed appropriately and rigorously? 

Reviewer #1: Yes

Reviewer #2: Yes

3. Have the authors made all data underlying the findings in their manuscript fully available?

Reviewer #1: Yes

Reviewer #2: Yes

4. Is the manuscript presented in an intelligible fashion and written in standard English?

Reviewer #1: Yes

Reviewer #2: Yes

5. Review Comments to the Author

Reviewer #1: There are very few systematic studies on ostrich eggshell beads, so these findings are a welcome addition to current research. I therefore suggest publication with minor changes. However, I recommend addressing a few issues in order to improve the paper.

Four main issues concern the analysis and interpretation of OES:

First, in the supplementary tables you mention the presence of OES in the DBSR stratigraphic aggregate. Considering the age of DBSR levels, the presence of one bead with possible ochre residues, three preforms and 44 ostrich eggshell fragments is very interesting. Why were they omitted from the analysis? Did you suspect their presence results from post-depositional mixing?

My second observation concerns the stages of bead manufacturing and interpretation of the material. All manufacturing stages are present but you did not mention that stages V to VII are much more frequent. This could be an interesting observation and should be taken into account in the interpretation of the material. Additionally, the fact that fragments are outnumbered by worked OES (lines 371-372) does not necessarily mean OES was used in a conservative manner. Maybe finished beads were produced at the site and then taken elsewhere.

Thirdly, the different types of usewear that were identified and recorded are not defined, resulting in a lack of clarity. I would therefore advise the authors to clearly define each type of usewear in the methods section.

Finally, the presence of ochre residues is mentioned throughout the text. How did you characterize these residues as ochre? You should specify in the methods section if it is a visual characterization, and which criteria were taken into account (colour, texture, etc). In order to use the term “ochre”, you should conduct chemical analyses to confirm the presence of goethite or hematite (or other iron oxides). As there are no elemental or mineralogical analyses to support this, I would advise using more general terms such as “red / yellow / coloured residues”.

Please find other minor comments and suggestions listed by page number below.

OES beads section

(1) Line 87. Perhaps it would be interesting not to restrict this section to beads and mention other ostrich eggshell occurrences. For instance, engraved ostrich eggshell from Diepkloof, dating back to 60 ka (Texier et al 2010) should be mentioned somewhere.

Context section: Ga-Mohana Hill North Rockshelter

(2) Lines 154-169. Although you cite Wilkins et al 2020, figures including the location, stratigraphy and plan of the excavated areas of the site would be welcome here.

(3) Line 170. In the supplementary tables there are pieces from the surface and the DBSR deposits. Do you omit those pieces in your study?

Methods

(4) Line 178. In the supplementary tables you added pieces from DBSR and surface layers. Here you should clearly explain why these pieces were omitted from the analysis (see main issues above).

(5) Line 183. Which criteria did you use to identify pigmentation? See comments above concerning the identification of ochre.

(6) Lines 184-185. See comments on the definitions of usewear above.

(7) Lines 186-189. It would be useful to the reader if you described each manufacturing stage rather than explaining them by groups (II-V and VI-VII)

(8) Lines 207-208. Data on the horizontal distribution does not appear in the supplementary tables.

Results

(9) Lines 230 and 234, and table 1. In supplementary table 2, there is one bead and one preform from DBAS. Do you mean DBGS? If it’s a typo, you should correct the numbers in the text and table 1 (20 beads and 10 preforms from DBGS).

(10) Line 234. “This layer”, do you mean DGBS? Again, you should clearly indicate in the methods section that you omit the pieces from DBSR and surface layers from the analysis although they appear in the supplementary tables.

(11) Lines 234-235. See comments on the stages of bead manufacturing above.

(12) Lines 241 and 260. See comments on the characterisation of ochre residues above.

(13) Lines 241 and 260. It would be interesting to show detailed photos of these residues as they are not clearly visible in figure 1.

(14) Line 244-246. How many preforms show these “pathway 1” features? Maybe a figure showing a couple of detailed photos of the “pathway 1” and “pathway 2” features could be useful here.

(15) Line 255. “The shape of each fragment was recorded following Miller”: this should be moved to the methods section.

(16) Line 259. “Each fragment was examined for residue using a hand lens”. This was already said in the methods section.

(17) Lines 262-265. The definition of “scratches” and criteria used to identify non-anthropogenic marks should be moved to the methods section.

(18) Lines 266-267. “OES colour is a useful tool for understanding whether a shell might have been exposed to heat and the temperature of the heat source […] OES becomes yellow, red, iridescent and grey when heated under oxidising conditions [48,53,62], while reducing conditions are more likely to produce blackening of the shell. Blackening of OES has not been replicated in experimental studies but is relatively common in the archaeological record [48,50,63]”: Perhaps this section should also be moved to the methods section.

(19) Lines 332-333. Maybe it would be better to specify “modern” ostrich here.

Discussion and conclusions

(20) Line 370. Again, the fact that stages V to VII are much more frequent could be an important observation.

(21) Lines 373-374. “Assuming that ostrich distribution has not changed drastically over the last 15 ka”: is there any reference that supports this?

(22) Lines 399-403. See above comments on the use of the term “ochre”.

Supplementary Tables:

(23) You could add a column with the horizontal provenance of the pieces. Do they all come from area A?

(24) The different types of usewear shown in the tables should be clearly defined in the methods section.

Please find below additional comments on the supplementary tables:

Supplementary table 2:

- Why is the bead ID column empty?

- Stratigraphic aggregates column: DBAS? This stratigraphic aggregate is not mentioned in the description of the stratigraphy, or in Wilkins et al 2020: I suppose it’s a typo? You probably mean DBGS.

- Width column: (mm) instead of (g)

- “Striae”, “smoothing”, “patina”, “chip”…: all use-wear types should be defined in the methods section. Perhaps locations should also be clearly explained.

- Staining column: what kind of staining is it when you indicate “yes” instead of “ochre”? You should be more specific here.

- Staining and comments columns: ochre was not identified using chemical analysis. You should either conduct elemental and mineralogical analysis to identify these residues as ochre or simply describe them without interpreting their composition (for example: red residues).

- Correct the Orton stage numbers: some of them are not in capital letters.

Supplementary table 3:

- Observations column: see comments on the use of the term “ochre” above.

- What do you mean by “surface modification”, as opposed to “human modification” and “non-human modifications”? This should appear in the material and methods section.

Reviewer #2: Thank you for allowing me to review the paper “Ostrich eggshell beads from Ga-Mohana Hill North Rockshelter, southern Kalahari, and the 2 implications for understanding social networks during Marine Isotope Stage 2.” This is a fantastic paper. The methodology is thorough, and the writing is clear. It is amazing to see the way that researchers are getting at use and manufacture of beads to look at population interactions. I really do not have any critiques. If the authors wish, they could tie the paper to the recent Nature article by Miller and Wang (2021). I recommend accepting the paper.

6. PLOS authors have the option to publish the peer review history of their article (what does this mean?). If published, this will include your full peer review and any attached files.

Reviewer #1: No

Reviewer #2: **Yes: **Jamie Hodgkins

---

## [Author Response · Author response to Decision Letter 0]

16 Mar 2022

Response to Reviewers

Manuscript PONE-D-21-22127

Dear Dr Spinapolice, 

Thank you for giving us the opportunity to submit a revised draft of the manuscript “Ostrich eggshell beads from Ga-Mohana Hill North Rockshelter, southern Kalahari, and the implications for understanding social networks during Marine Isotope Stage 2.” for publication in PLOS ONE. 

We thank the reviewers for their constructive suggestions, which we addressed in this revised submission, and we believe that the manuscript is now substantially improved. Below is a detailed point-by-point response to each reviewer's comment.

Editors Comments/Journal Requirements:

Authors' response: We have made sure that the manuscript meets the requirements.

 2. In your manuscript, please provide additional information regarding the specimens used in your study. Ensure that you have reported specimen numbers and complete repository information, including museum name and geographic location. If permits were required, please ensure that you have provided details for all permits that were obtained, including the full name of the issuing authority, and add the following statement: All necessary permits were obtained for the described study, which complied with all relevant regulations.' If no permits were required, please include the following statement: 'No permits were required for the described study, which complied with all relevant regulations.'

Authors' response: Supplementary Tables 2 and 3 have the specimen number, we have renamed the columns from ‘finds’ to ‘specimen number’ to indicate this. We have included more information about where the specimens are located in the methods section. And included this sentence to the methods section: “All necessary permits were obtained for the described study, which complied with all relevant regulations. Excavations at Ga-Mohana Hill were approved by the South African Heritage Resources Agency under permit ID 2194. All specimen numbers relevant to this study are provided in Supplementary Tables 2 and 3 . These specimens are currently housed in the Archaeology Department at the University of Cape Town and they will be permanently curated by the McGregor Museum, Kimberley, Northern Cape, South Africa.”

Authors' response: Ethics approval is not relevant to this study as it did not involve human subjects

 4. We note that Figures 3, 4, 5, 6, and 7 in your submission contain map images which may be copyrighted. All PLOS content is published under the Creative Commons Attribution License (CC BY 4.0), which means that the manuscript, images, and Supporting Information files will be freely available online, and any third party is permitted to access, download, copy, distribute, and use these materials in any way, even commercially, with proper attribution. For these reasons, we cannot publish previously copyrighted maps or satellite images created using proprietary data, such as Google software (Google Maps, Street View, and Earth).

Authors' response: The data used to create the maps is from 2 public domain sources USGS EROS (Earth Resources Observatory and Science (EROS) Center) (public domain): http://eros.usgs.gov/# and Natural Earth (public domain): http://www.naturalearthdata.com/. The data for ostrich distribution is adapted from the South African Bird Atlas Project 2 (SABAP2) acessed at (http: //sabap2.birdmap.africa/) which is copyright under CC BY 4.0. In the text we attribute the SABAP2 data, but we have added to the figure captions to clarify this.

Authors' response: We have added the following references to the reference list, based on citations that were added to address reviewers comments. Miller and Wang (2022); Thomas and Shaw (2002).

Reviewer 1 (anonymous) 

There are very few systematic studies on ostrich eggshell beads, so these findings are a welcome addition to current research. I therefore suggest publication with minor changes. However, I recommend addressing a few issues in order to improve the paper. 

Authors' response: Thank you!

 1. In the supplementary tables you mention the presence of OES in the DBSR stratigraphic aggregate. Considering the age of DBSR levels, the presence of one bead with possible ochre residues, three preforms and 44 ostrich eggshell fragments is very interesting. Why were they omitted from the analysis? Did you suspect their presence results from post-depositional mixing? 

Authors' response: This paper is focused only on reporting the DBGS OES. A small number of preforms and a bead are associated with the DBSR deposit, however, we need to investigate this association further. At this point we are too uncertain about this association to report such an early age for OES beads. We're worried that the small number of tiny artefacts may have fallen from the walls during excavation - further excavations will help us resolve this. The DBSR beads and OES have now been excluded from the supp files and it is made clear now that only the DBGS is being reported here. The DBAS is a strag agg in the LSA deposits in the south shelter (GHS), which is still pending publication, and should have been excluded from the supp file.

 2. My second observation concerns the stages of bead manufacturing and interpretation of the material. All manufacturing stages are present but you did not mention that stages V to VII are much more frequent. This could be an interesting observation and should be taken into account in the interpretation of the material. Additionally, the fact that fragments are outnumbered by worked OES (lines 371-372) does not necessarily mean OES was used in a conservative manner. Maybe finished beads were produced at the site and then taken elsewhere. 

Authors' response: We have added to the Bead Manufacture section of the discussion to further discuss the higher frequency of stages V to VII and added to the sentence about OES being used in a conservative manner.

 3. Thirdly, the different types of usewear that were identified and recorded are not defined, resulting in a lack of clarity. I would therefore advise the authors to clearly define each type of usewear in the methods section. 

Authors' response: We have defined the usewear types in the methods section.

 4. Finally, the presence of ochre residues is mentioned throughout the text. How did you characterize these residues as ochre? You should specify in the methods section if it is a visual characterization, and which criteria were taken into account (colour, texture, etc). In order to use the term “ochre”, you should conduct chemical analyses to confirm the presence of goethite or hematite (or other iron oxides). As there are no elemental or mineralogical analyses to support this, I would advise using more general terms such as “red / yellow / coloured residues”. 

Authors' response: We have added a sentence specifying that this was a visual characterisation, and changed all mentions of ochre to red/ yellow coloured residue.

OES beads section 

 5. Line 87. Perhaps it would be interesting not to restrict this section to beads and mention other ostrich eggshell occurrences. For instance, engraved ostrich eggshell from Diepkloof, dating back to 60 ka (Texier et al 2010) should be mentioned somewhere. 

Authors' response: We have added the Texier et al. 2010 reference in this section. However we have decided not to include more information on other oes fragment occurrences as they are generally only reported if they have been modified (decorated), while the oes fragments we report on are unmodified.

Context section: Ga-Mohana Hill North Rockshelter 

 6. Lines 154-169. Although you cite Wilkins et al 2020, figures including the location, stratigraphy and plan of the excavated areas of the site would be welcome here. 

Authors' response: We have included a figure (figure 1) to show the location and stratigraphy of the site.

 7. Line 170. In the supplementary tables there are pieces from the surface and the DBSR deposits. Do you omit those pieces in your study? 

Authors' response: Yes, the only pieces included in the study are from DBGS stratigraphic aggregate. This is mentioned in the last paragraph of the Ga-Mohana Hill North Rockshelter section as a lead in for the Methods section. We have also clarified this in the Methods section.

Methods

 8. Line 178. In the supplementary tables you added pieces from DBSR and surface layers. Here you should clearly explain why these pieces were omitted from the analysis (see main issues above). 

Authors' response: Added this sentence to clarify: “The study was limited to analysis of OES assemblage from the DBGS stratigraphic aggregate layer in order to investigate and compare MIS 2 sites with OES assemblages across southern Africa.”

 9. Line 183. Which criteria did you use to identify pigmentation? See comments above concerning the identification of ochre. 

Authors' response: Added this sentence: “Pigmentation was classified visually based on colour and texture.“

 10. Lines 184-185. See comments on the definitions of usewear above. 

Authors' response: Have added more information about the usewear classification in the methods section.

 11. Lines 186-189. It would be useful to the reader if you described each manufacturing stage rather than explaining them by groups (II-V and VI-VII) 

Authors' response: We have included descriptions of each manufacturing stage.

 12. Lines 207-208. Data on the horizontal distribution does not appear in the supplementary tables. 

Authors' response: This data is already included in figure 3.

Results

 13. Lines 230 and 234, and table 1. In supplementary table 2, there is one bead and one preform from DBAS. Do you mean DBGS? If it’s a typo, you should correct the numbers in the text and table 1 (20 beads and 10 preforms from DBGS). 

Authors' response: The DBAS is a strag agg in the LSA deposits in the south shelter (GHS), which is still pending publication, and should have been excluded from the supp file.

 14. Line 234. “This layer”, do you mean DGBS? Again, you should clearly indicate in the methods section that you omit the pieces from DBSR and surface layers from the analysis although they appear in the supplementary tables. 

Authors' response: Changed “This layer” to DBGS, and have corrected the methods section to clarify the analysis only includes DBGS

 15. Lines 234-235. See comments on the stages of bead manufacturing above. 

Authors' response: Have added more detail on manufacturing stages above

 16. Lines 241 and 260. See comments on the characterisation of ochre residues above.

Authors' response: changed “ochreous residue” to “red coloured residue

 17. Lines 241 and 260. It would be interesting to show detailed photos of these residues as they are not clearly visible in figure 1. 

Authors' response: This is unfortunately not possible at the moment as the beads are stored in Cape Town and none of the authors are in South Africa at the moment.

 18. Line 244-246. How many preforms show these “pathway 1” features? Maybe a figure showing a couple of detailed photos of the “pathway 1” and “pathway 2” features could be useful here. 

Authors' response: Unfortunately not possible as explained above

 19. Line 255. “The shape of each fragment was recorded following Miller”: this should be moved to the methods section. 

Authors' response: Moved to the methods section

 20. Line 259. “Each fragment was examined for residue using a hand lens”. This was already said in the methods section. 

Authors' response: Removed from the methods section

 21. Lines 262-265. The definition of “scratches” and criteria used to identify non-anthropogenic marks should be moved to the methods section. 

Authors' response: Moved to the methods section

 22. Lines 266-267. “OES colour is a useful tool for understanding whether a shell might have been exposed to heat and the temperature of the heat source […] OES becomes yellow, red, iridescent and grey when heated under oxidising conditions [48,53,62], while reducing conditions are more likely to produce blackening of the shell. Blackening of OES has not been replicated in experimental studies but is relatively common in the archaeological record [48,50,63]”: Perhaps this section should also be moved to the methods section. 

Authors' response: Moved to the methods section.

 23. Lines 332-333. Maybe it would be better to specify “modern” ostrich here. 

Authors' response: Have added modern to the sentence to clarify.

Discussion and conclusion

 24. Line 370. Again, the fact that stages V to VII are much more frequent could be an important observation. 

Authors' response: Have added this sentence to highlight the higher proportion of beads to preforms. “There is however a higher proportion of beads compared to preforms, which could indicate that drilling and grinding of beads were conducted in different areas of the rockshelter.”

 25. Lines 373-374. “Assuming that ostrich distribution has not changed drastically over the last 15 ka”: is there any reference that supports this? 

Authors' response: Unfortunately there is not much data on past ostrich distribution, we have changed this sentence to “Although data on changes in ostrich distribution in the late Quaternary are limited, if we assume that ostrich distribution has been relatively consistent across wet and dry phases over the last 15 ka (Thomas and Shaw 2002), then ostrich eggs and their shells would have been regularly encountered.”

 26. Lines 399-403. See above comments on the use of the term “ochre”. 

Authors' response: Have removed ochre for description of residues on OES and replaced with red residues.

Supplementary Tables

 27. You could add a column with the horizontal provenance of the pieces. Do they all come from area A? 

Authors' response: The horizontal provenience is shown in figure 2 (the main square is a view from above while each of the two side panels look north and east respectively). They all come from Area A – We have added the figure (figure 1) showing the rockshelter and excavated areas which might help clear this up.

 28. 4) The different types of usewear shown in the tables should be clearly defined in the methods section. 

Authors' response: Added this sentence to the methods section to define use-wear “Striations are defined as randomly oriented short striations, while facets are depressions on the surface of the bead.”

 29. Why is the bead ID column empty? 

Authors' response: We have removed this column – not applicable for this dataset.

 30. Stratigraphic aggregates column: DBAS? This stratigraphic aggregate is not mentioned in the description of the stratigraphy, or in Wilkins et al 2020: I suppose it’s a typo? You probably mean DBGS. 

Authors' response: The DBAS is a strag agg in the LSA deposits in the south shelter (GHS), which is still pending publication, and should have been excluded from the supp file.

 31. Width column: (mm) instead of (g) 

Authors' response: Fixed.

 32. “Striae”, “smoothing”, “patina”, “chip”…: all use-wear types should be defined in the methods section. Perhaps locations should also be clearly explained. 

Authors' response: We have included a sentence listing all of the attributes that were recorded along with a reference to the publication where more detailed explanations can be found. Including descriptions of each of these attributes would be very wordy and many of the attribute have not been studied further in this publication.

 33. Staining column: what kind of staining is it when you indicate “yes” instead of “ochre”? You should be more specific here. 

Authors' response: Have updated this to be more specific, yes was referring to brown staining on the OES.

 34. Staining and comments columns: ochre was not identified using chemical analysis. You should either conduct elemental and mineralogical analysis to identify these residues as ochre or simply describe them without interpreting their composition (for example: red residues). 

Authors' response: We have changed all references to ochre on the beads or fragments to red residue.

 35. Correct the Orton stage numbers: some of them are not in capital letters. 

Authors' response: corrected.

 36. Observations column: see comments on the use of the term “ochre” above. 

Authors' response: Have replaced ochre with red residue throughout.

 37. What do you mean by “surface modification”, as opposed to “human modification” and “non-human modifications”? This should appear in the material and methods section. 

Authors' response: Surface modifications are any markings on the surface that we could not confidently ascribe as either taphonomic or anthropogenic. We have edited the column headings in the supplementary table to better describe them. As well as adding a description of surface modification to the methods section.

Reviewer 2

Thank you for allowing me to review the paper “Ostrich eggshell beads from Ga-Mohana Hill North Rockshelter, southern Kalahari, and the 2 implications for understanding social networks during Marine Isotope Stage 2.” This is a fantastic paper. The methodology is thorough, and the writing is clear. It is amazing to see the way that researchers are getting at use and manufacture of beads to look at population interactions. I really do not have any critiques. If the authors wish, they could tie the paper to the recent Nature article by Miller and Wang (2021). I recommend accepting the paper. 

Authors' response: Thank you! We have added the Miller and Wang reference to the introduction and discussion.

---

## [Decision Letter · Decision Letter 1]

12 May 2022

Ostrich eggshell beads from Ga-Mohana Hill North Rockshelter, southern Kalahari, and the implications for understanding social networks during Marine Isotope Stage 2.

PONE-D-21-22127R1

Dear Dr. Hatton,

We’re pleased to inform you that your manuscript has been judged scientifically suitable for publication and will be formally accepted for publication once it meets all outstanding technical requirements.

Kind regards,

Enza Elena Spinapolice, Ph.D

Academic Editor

PLOS ONE

Additional Editor Comments (optional):

Reviewers' comments:

Reviewer's Responses to Questions

**Comments to the Author**

1. If the authors have adequately addressed your comments raised in a previous round of review and you feel that this manuscript is now acceptable for publication, you may indicate that here to bypass the “Comments to the Author” section, enter your conflict of interest statement in the “Confidential to Editor” section, and submit your "Accept" recommendation.

Reviewer #2: All comments have been addressed

2. Is the manuscript technically sound, and do the data support the conclusions?

Reviewer #2: Yes

3. Has the statistical analysis been performed appropriately and rigorously? 

Reviewer #2: Yes

4. Have the authors made all data underlying the findings in their manuscript fully available?

Reviewer #2: Yes

5. Is the manuscript presented in an intelligible fashion and written in standard English?

Reviewer #2: Yes

6. Review Comments to the Author

Reviewer #2: I reviewed this manuscript very positively the first time, and my opinion has not changed. This is a scientifically sound article, that adds insight to the field.

7. PLOS authors have the option to publish the peer review history of their article (what does this mean?). If published, this will include your full peer review and any attached files.

Reviewer #2: **Yes: **Jamie Hodgkins

---

## [Editor Report · Acceptance letter]

23 May 2022

PONE-D-21-22127R1 

Ostrich eggshell beads from Ga-Mohana Hill North Rockshelter, southern Kalahari, and the implications for understanding social networks during Marine Isotope Stage 2 

Dear Dr. Hatton:

I'm pleased to inform you that your manuscript has been deemed suitable for publication in PLOS ONE. Congratulations! Your manuscript is now with our production department. 

Kind regards, 

on behalf of

Dr. Enza Elena Spinapolice 

Academic Editor

PLOS ONE